# Applications of High-Resolution Imaging for Open Field Container Nursery Counting

**Ying She [1], Reza Ehsani [2,\*], James Robbins [3], Josué Nahún Leiva [3] and Jim Owen [4]**

[1]   The Climate Corporation, San Francisco, CA 94103, USA; yingshe1015@gmail.com
[2]   School of Engineering, University of California, Merced, CA 95343, USA
[3]   Department of Horticulture, University of Arkansas, Fayetteville, AR 72701, USA; jrobbin@uark.edu (J.R.);
      nahunleiva@gmail.com (J.N.L.)
[4]   School of Plant and Environmental Sciences, Virginia Polytechnic Institute and State University,
      Virginia Beach, VA 23455, USA; jim.owen@vt.edu
\*   Correspondence: rehsani@ucmerced.edu; Tel.: +1-863-398-9585

**Abstract:** Frequent inventory data of container nurseries is needed by growers to ensure proper management and marketing strategies. In this paper, inventory data are estimated from aerial images. Since there are thousands of nursery species, it is difficult to find a generic classification algorithm for all cases. In this paper, the development of classification methods was confined to three representative categories: green foliage, yellow foliage, and flowering plants. Vegetation index thresholding and the support vector machine (SVM) were used for classification. Classification accuracies greater than 97% were obtained for each case. Based on the classification results, an algorithm based on canopy area mapping was built for counting. The effects of flight altitude, container spacing, and ground cover type were evaluated. Results showed that container spacing and interaction of container spacing with ground cover type have a significant effect on counting accuracy. To mimic the practical shipping and moving process, incomplete blocks with different voids were created. Results showed that the more plants removed from the block, the higher the accuracy. The developed algorithm was tested on irregular- or regular-shaped plants and plants with and without flowers to test the stability of the algorithm, and accuracies greater than 94% were obtained.

**Keywords:** image processing; container nursery; inventory management; counting; overlap separation

## 1. Introduction

Container nurseries are an example of compact and intensive agricultural production. Generally, one hectare of land may hold 80,000 to 600,000 containers [1]. Management of containerized crop production is labor-intensive and time consuming, especially for large nursery production areas. In some nurseries, the inventory is extrapolated from a count of only a portion of the crop due to the time involved in manually counting each plant [2]. One advancement in automating inventory management is the use of radio frequency identification (RFID). However, the RFID system only works when within close proximity (small read range) to avoid transmission error [3]. Robbins et al. [4] mentioned that the primary challenge of applying a RFID system to nursery production is to match the proper tag with different production systems and environmental conditions which makes the RFID system less adaptive.

With the rapid growth of unmanned aerial vehicles (UAV) and their wide application in agriculture [5–10], research focusing on container-grown nursery inventory management with aerial-based approaches has emerged in response to the aforementioned limitations of ground-based

approaches. To the best knowledge of the authors, only one research trial has been reported using aerial imagery to count container-grown nursery plants. Leiva [4] used Feature Analyst (Textron Systems, Providence, RI), an object-based image analysis software, to count open-field container nursery plants based on images collected by UAV. However, this method requires subjective user input and parameter settings, which could be a source of potential error. Moreover, directly applying methods developed via third-party software provides little flexibility and embedded cost to allow researchers or growers to develop customized methods for nursery counting.

Although there is limited research focusing on application of imaging for container-grown nursery counting, there has been a long history of its broad application on other counting tasks in agriculture, such as fruit and tree counting. Wang et al. [11] used saturation, hue, and local maximum of specular reflections in four directions to detect apple pixels and then split touching apples and merged the occluded apple based on the estimate diameter. Payne at al. [12] segmented images into mango fruits and background with color information in red-green-blue (RGB), YCbCr space and texture information. Then a lower and an upper limit were applied to block size and subsequently estimate mango number. An automated kiwifruit counting technique was proposed so that kiwifruit was firstly extracted by minimum distance classification in L*a*b color space, and then a regression relationship was utilized to estimate the true fruit count [13]. Nuske et al. [14] detected grape locations using radial symmetry transformation and removed false position locations based on cluster size. Annamalai [15] developed a citrus yield estimate system based on color images. Citrus fruits were extracted from background with hue and saturation information and then counting was conducted on the segmentation result. Kurtulmus et al. [16] detected green citrus fruit with 'eigenfruit', color, and circular Gabor texture features. For counting citrus fruits, blob analysis was performed to merge multiple detections for the same fruit. A similar merge scheme was used to estimate the number of apple fruits when partially occluded [17]. Gnädinger el al. [18] adopted a decorrelation stretch contrast enhancement procedure to enhance color different which enabled detection of the center of maize plants for counting purposes. An apple tree counting method was developed by Quirós et al. [19] which applied filters based on the size and location of the polygons to rasterized the image in order to isolate apple plants.

All the aforementioned counting methods included a two-step process in common: (1) segment plants/fruits pixels from background to rasterize images; and (2) development of the counting scheme. The segmentation is application-dependent as different segmentation methods have been used to detect and distinguish leaves, fruits, trees, and weeds. The development of the counting scheme is also application-dependent because plants and fruits present different color characteristics and they have different degrees of interaction with adjacent plants/fruits. For example, Wang et al. [11] dealt with the touching fruits with the hypothesis that only two fruits are connected. In container-nursery production, severe canopy interactions between plants are usually expected. Although the details may vary across different counting tasks, the common two-step pipeline for counting will be applied in our work with development of segmentation and counting scheme applicable to container-grown nursery counting. Different from the counting schemes based on object detection (shape, color, size, etc.) in the previous works, a novel counting scheme based on the plant canopy area will be developed in MATLAB 2013b.

In addition to the algorithm design, the effects of some important factors on the algorithm were evaluated herein. The first factor is altitude. By capturing images at different altitudes with a fixed camera, the effect of image resolution can be evaluated. The second factor is container plant spacing or density. Thirdly, the effect of the ground cover on counting accuracy was evaluated. Since container nursery plants are typically produced on black fabric and gravel, the accuracy was compared on these two backgrounds. Fourthly, since the number of containers in a fixed block is changing due to plant movement or shipping, different missing scenarios were evaluated. Fifthly, there are some variations in the plants themselves. In this study, the variations were confined to flower blooming timings and shape. The performance of the counting scheme was evaluated on representative irregular and regular shaped plants as well as plants with and without flowers.

In summary, the objectives of this study were: (1) to build robust classifiers in order to separate container-grown nursery plants from the production background for selected, representative plant types; (2) to develop a split scheme for severely touching canopies; (3) to evaluate the effect of flight altitude (image resolution), container spacing/canopy density and ground cover (black fabric and gravel) on the developed algorithm; (4) to explore the stability of the algorithm, specifically, to evaluate the performance on incomplete crop blocks and to evaluate whether the counting algorithm is dependent on shape or growing stage.

## 2. Materials and Methods

### 2.1. Data Collection

An 800AJ articulated boom (JLG Industries, Inc., McConnellsburg, PA, USA) as shown in Figure 1 was used to collect Dataset 1 and Dataset 2. We started with a boom which is able to remain still at a fixed height for a long period to allow us to rearrange the plants with different spacing. The experiment was conducted on 13 and 14 November 2012 at the Citrus Research and Education Center (University of Florida, FL, USA). The available height range of the boom was 9.14 m to 24.38 m (30 to 80 ft.). Sensors were mounted to an aluminum pole that extended 2 m horizontally beyond the bucket of the boom. Images were taken after a proper height (altitude) and location was obtained by adjusting the extendable boom. A plumb line was used to locate the center of the plant region to guarantee that the image had complete coverage of all the plants. In this research, plants were selected to represent the general characteristics of ornamental plants and the species selected are grown in large numbers in commercial production.

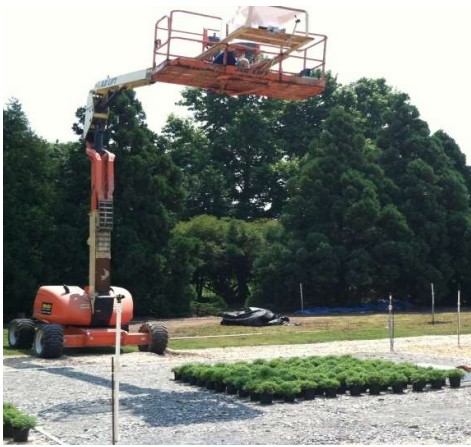

**Figure 1.** Articulated boom platform.

### 2.1.1. Dataset 1. Evaluation of the Effect of Image Resolution and Plant Spacing of Perennial Peanut on Plant Count

The experimental field was a block of 100 containers spaced in a $10 \times 10$ container grid on a black fabric cover. The perennial peanut (*Arachis glabrata*) is a fast-growing crop that has a uniform/regular outline to its canopy. The containers used were the Nursery Supplies® blow molded C200 pots with the following specifications: volume, 2.03 L; top diameter, 15.24 cm, height, 15.24 cm. Images were taken at the heights off the ground of 9, 12, 15, and 18 m (30, 40, 50, and 60 ft.). In addition, container spacing of 0, 3, 8, 13, 18, and 23 cm (0, 1, 3, 5, 7 and 9 inches) which corresponds to a plant density of 43.0, 31.6, 19.1, 12.8, 9.2, 6.9 plants/m² was evaluated. Twenty-four treatments (4 heights $\times$ 6 plant densities) were evaluated with four replicates each. In order to validate the results, 20 plants were switched with 20 other plants inside the block randomly after photographing each replicate. The center location of the $10 \times 10$ block was manually marked. Before each image was taken, the boom operator used a plumb line to check whether the camera was directly over the center point of a block in order to

make sure the image did not deviate. This manual process is to make sure each single image could capture the whole experimental block. This step is not necessary for the UAV platform since images could be stitched together to give a full view. Although this process was undertaken, some images still deviated due to camera orientation. The deviation reduced the available number of combined treatments to 17. The camera used in this experiment was a Canon EOS 5D Mark II with a resolution of 5616 × 3744 pixels.

### 2.1.2. Dataset 2. Evaluation of the Effect of a Partially Filled Block on Plant Count

The plant species, container size, and articulated boom used for this dataset are the same as those described in Dataset 1. The platform was fixed at 12 m (40 feet), and container spacing was set at 8 cm (3 in). Twenty plants were randomly moved out from the complete 10 × 10 block each cycle until only 20 plants remained. In total, there were four treatments in which 80, 60, 40, and 20 plants remained randomly spaced within the initial area of the 10 × 10 block. For each treatment, the plants were shuffled randomly three times. In each shuffling, plants were switched randomly so that not only the locations of the plants but also the voids were ever-changing. This setting was to mimic the incomplete blocks with voids due to the movement or shipping of container-grown nurseries in practical nursery production.

### 2.1.3. Dataset 3. Exploring the Effect of Flight Altitude, Plant Spacing and Ground Cover for Fire Chief$^{TM}$ Arborvitae

Container-grown Fire Chief$^{TM}$ arborvitae (*Thuja occidentalis* L.), grown in #3 black polyethylene containers (height: 23.5 cm, top diameter: 26.5 cm, and bottom diameter: 23.0 cm) were spaced in staggered rows to achieve three canopy separation treatments: 5 cm between canopy edges (completely separated), 0 cm between canopy edges (edge touching), and −5 cm between canopy edges (5 cm canopy edge overlap) which correspond to a plant density of 8.8, 11.9, and 17.6 plants/m$^2$, respectively. For the species selected, the crop density is reflective of actual scenarios in commercial nursery production. In this experiment, plant densities were limited to three representative settings. Three treatment sets were replicated three times in a randomized complete block design (RCBD) for a total of nine sets of plants as shown in Figure 2. For each set, 64 plants (8 × 8) were placed on gravel. One set with a canopy separation of −5 cm only had 56 plants (7 × 8) due to a miscounting when the experiment was set up. The nine sets were placed 3 × 3. Four fully separated plants were placed outside each of the nine sets and were used to train the open-source MATLAB algorithm. The same canopy treatment was repeated on black polypropylene fabric ground cover (Lumite, Inc., Alto, GA, USA) by repositioning the plants on the fabric cover.

A multi-rotor UAV mounted with a Sony NEX-5n 16.1 megapixel color digital frame camera with an 18–55 mm lens was used. To evaluate the effect of flight height, three flights of 6, 12, and 22 m were made on 13 July 2013 (gravel) and 14 July 2013 (fabric) at Greenleaf Nursery, Park Hill, OK, USA. Each flight was carried out in a zigzag pattern to cover all nine sets as indicated by the gray arrow in Figure 2. The flight at each of the three altitudes was executed two times with the same flight path. The first flight was referred to as run 1 and second as run 2. The sky was sunny when images were taken. Flights for the gravel background were conducted between 07:40 a.m. and 10:30 a.m. with a wind speed from 0 to 9 km/h while flights using a fabric background were conducted between 08:00 a.m. and 9:30 a.m. with a wind speed from 0 to 11 km/h.

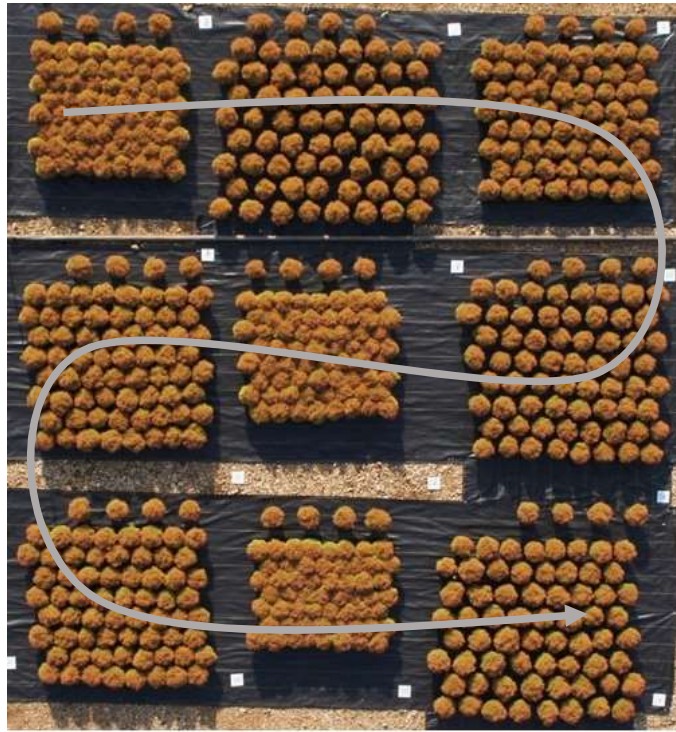

**Figure 2.** Aerial image of the experimental layout for container arborvitae.2.1.4 Dataset 4. Exploring the stability of counting methodology with different shapes and in different growing stages.

Two species of Juniper (*Juniperus chinensis* 'Sea Green' and *Juniperus horizontalis* 'Plumosa Compacta') were selected to evaluate the effect of shape on plant counts. 'Sea Green' was chosen as the plant with an 'irregular' canopy shape while 'Plumosa Compacta' was chosen as the 'regular' shape. In addition, Drift® rose (*Rosa* sp. 'Meidrifora') was selected to evaluate the effect of presence of flowers. There are two treatments: rose plants with flowers and rose plants without flowers. For roses without flowers, flowers were manually removed. In this experiment, the manual removal was simply to mimic the growth stage for plants without flowers; growers need not perform this step when applying our proposed counting method.

For each treatment in canopy shape and flowering, 64 plants (8 × 8) were placed with a 5 cm canopy separation on black polypropylene fabric ground cover. The two treatment sets in both experiments were replicated five times in RCBD. Four plants of treatment 1 and four of treatment 2 were placed outside all sets for the purpose of developing and training an open-source MATLAB algorithm. A Bil-Jax 3632T boom lift (Haulotte Group, Archbold, OH, USA) fixed at 12 m above ground level was used to collect images on 13 November 2013 at Greenleaf Nursery, Park Hill, OK, USA. The sensor used in this experiment was a Sony Alpha NEX-7 (Sony Corporation of America IR, San Diego, CA, USA), 24.3 megapixels color digital frame camera, with an 18–55 mm lens.

*2.2. Image Segmentation*

To develop and evaluate different classification methods for different cases, three representative cases were selected: perennial peanut (green), arborvitae (yellow) and rose (green with flowers). A favorable factor for image segmentation of nursery plants is in the USA, open-air container-grown nursery crops are routinely placed on black polypropylene fabric for erosion control and prevention of weeds under and between the containers [20]. The black cover creates a uniform mask, thus simplifying the background. Due to the large contrast between plants and uniform black background, threshold based segmentation methods are applied. Image pre-processing was performed before image segmentation. A 3 × 3 median filter was used to remove noise while preserving edge information. To increase the image contrast, the upper 1% and lower 1% of gray scale data was mapped to 1 and 0,

respectively. Data between the upper and lower boundary were mapped to (0, 1). Image normalization is an important step to make the threshold-based method reliable across images under different light conditions. Due to the relatively constant light condition during image collection, normalization was not performed. Image post-processing including erosion and dilation were performed to remove small false identified parts.

### 2.2.1. Green Plant Segmentation

To highlight the green component, the gray scale component excess green index (ExG = (2 × g − r − b) [21,22] was applied. ExG of black fabric will be approximately 0 while ExG of green plants will be substantially greater than 0. A simple two-class classification scheme based on a threshold is shown below:

$$class = \begin{cases} 1 & component \: |value < threshold \\ 2 & component \: |value \geq threshold \end{cases}$$

Five thousand pixels were randomly chosen for each class: black fabric and green plants. The distributions of (2 × g − r − b) of the background and green plants are shown in Figure 3. It can be clearly seen that there is no overlap between the histograms of the two classes. The ExG of green plants is much higher than that of the background. A global threshold (0.05) was then generated to segment the image.

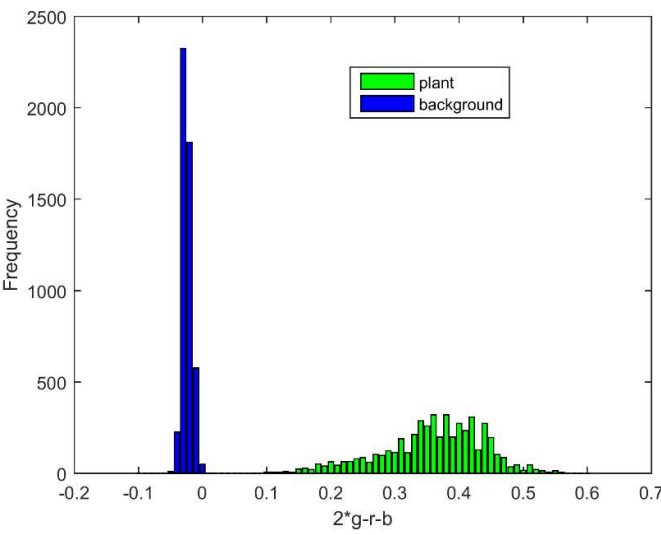

**Figure 3.** Histograms of (2 × g − r − b) for plants and background.

### 2.2.2. Yellow Plant Segmentation

For black fabric, the reflectance at the red, green, and blue bands is close to zero. For yellow objects, there is a large difference in the reflectance of the red and blue bands. We took the normalized index $\frac{(r-b)}{(r+b)}$ to segment yellow objects from the background. When images were taken, shadows occurred on the background. The distribution of the normalized index of yellow plants, background, and shadow are shown in Figure 4. The figure shows that the histogram of shadow overlaps with both of plant and background. However, the plants and background are completely separated. Based on the histogram, the setting threshold in the range of [0.05, 0.3] could aid in removal of all background pixels and a portion of the shadow pixels. The next step was to separate the plants from the remaining shadow pixels. Since the shadow pixels are dark in color, a novel index 3 − (r + b + g) − k × (|r − b| + |g − b| + |r − g|) was taken to segment the dark pixels from other pixels. This index was chosen because both the r, g, b values and the absolute difference between each of the two components are small for dark pixels, facilitating separation of dark from plant pixels. The value of k was set to 10 via a trial and error process. The histograms of 3 − (r + b + g) − 10 × (|r − b| + |g − b| + |r − g|) of

the background and plant pixels are shown in Figure 5. The application of this index allowed shadow pixels to be extracted. By overlaying the shadow pixels onto the image, which contains plants and part of a shadow, plant pixels could be successfully extracted. The processing of an example image is shown in Figure 6.

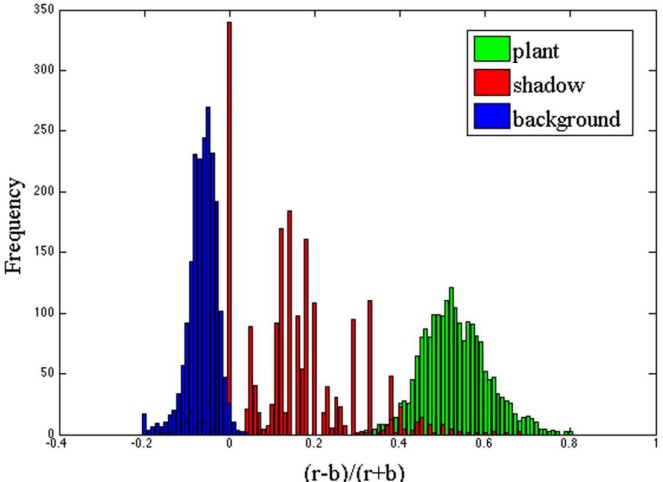

**Figure 4.** Histograms of $\frac{(r-b)}{(r+b)}$ for plants, shadow and background.

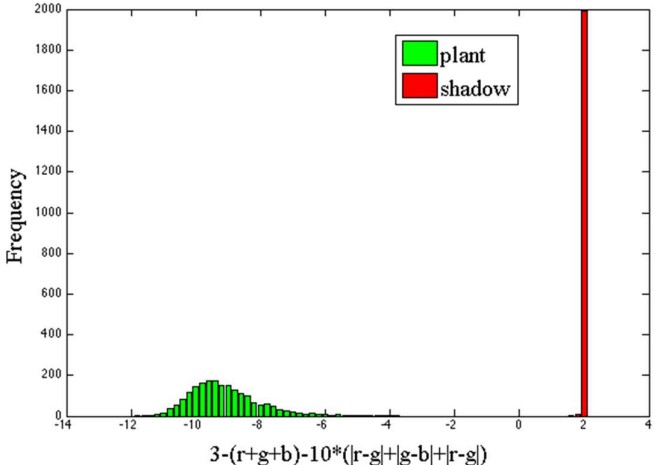

**Figure 5.** Histograms of $3 - (r + b + g) - 10 \times (|r - b| + |g - b| + |r - g|)$ for plants and background.

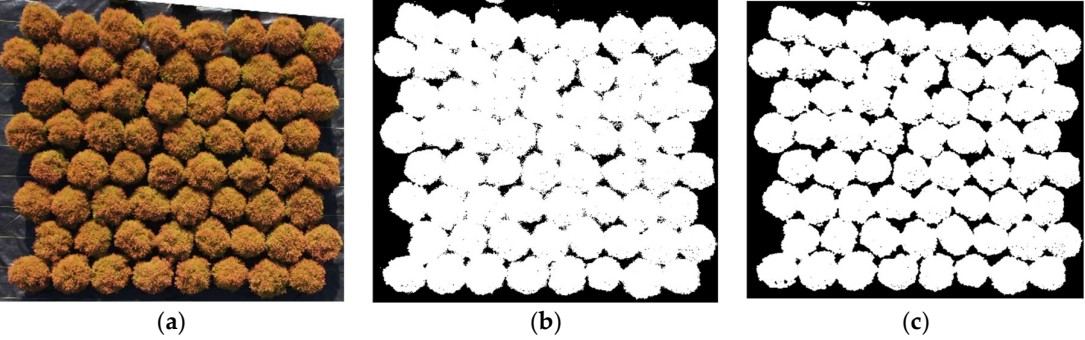

**Figure 6.** Overview of the plant extraction process: (**a**) original image; (**b**) image extracted by index $\frac{r-b}{r+b}$ (**c**) removing remaining shadow pixels between plants.

### 2.2.3. Flower Plant Segmentation

For rose plants with flowers, the region of interest is a mixture of red flowers and green leaves. Support vector machine (SVM) [23] was used for image classification. The first step in SVM is to build training data which is composed of feature vectors together with known labels. The goal of SVM is to predict the unknown labels of the test data given only the feature vectors of the test data. Training data are represented as instance-label pairs $(x_i, y_i), i = 1, 2, \ldots, l; x_i \in R_n$ (n is the feature dimension); $y_i \in \{-1, 1\}$. The basic idea of SVM is to find the optimal maximum-margin hyperplane represented by Equation (1) which could linearly separate the points labeled as –1 from the points labeled as 1. Input vectors that lie on the decision boundaries are called support vectors.

$$t_i = w * x_i + b \tag{1}$$

where:

$t_i$ is the separating hyperplane equation,

w is the weight vector,

$x_i$ is the feature vector,

b is the bias.

The margin between the two hyperplanes is $\frac{2}{\|w\|}$. With the condition that:

for all points which are labeled as $-1$ $(y_i = -1)$, $t_i \leq -1$;

for all points which are labeled as 1 $(y_i = 1)$, $t_i \geq -1$

These two conditions are combined into $t_i * y_i \geq 1$, which indicates $y_i * (w * x_i + b) \geq 1$. So, the optimization function for SVM is argmin $\frac{\|w\|}{2}$, subject to (for any $i = 1, 2, \ldots, k$) $y_i * (w * x_i + b) \geq 1$.

This is a quadratic optimization problem and could be solved using the Lagrange multiplier approach [24].

The LIBSVM tool [25] was chosen to segment nursery plants from the background. LIBSVM is an interface for SVM classification and regression. A LIBSVM MATLAB package was used for classification purposes. Firstly, 10,000 pixels of pure foreground (leaves and flowers) and 10,000 pixels of pure background were selected manually. The foreground pixels were labeled as 1 while background pixels were labeled as $-1$. A three-dimensional feature {r, g, b} space was created for the two categories. Eighty percent (8000 pixels) of each category was used for training the model and 20% (2000 pixels) of each category was used for testing purpose. The scatter plot of feature vectors of the background and the foreground can be visualized in Figure 7. Before applying SVM, it is recommended to scale the attribute in the feature vector to $(-1,1)$ or $(0,1)$ to avoid attributes with a large number dominating attributes with a small number [26]. In this research, the value of R, G and B is already in the range of (0,1). Therefore, scaling was not needed.

From the feature space, it is observed that separation of these two categories with a linear classifier is not possible. Hence, kernel function was applied to transform the original data into a higher dimension space. Since the feature dimension is much smaller than training size, the RBF kernel is recommended. The RBF kernel is represented as $e^{-\gamma|\mu - \nu|^2}$. In order to build the SVM classifier, two parameters including (1) the penalty parameter C and (2) the kernel parameter $\gamma$ for the RBF kernel must be decided. The penalty parameter C is a trade-off between the margin width of two categories and the number of outliers. To choose the best parameters, a 10-fold cross validation was applied. A grid search approach was used to find the $(C, \gamma)$ pair with the best cross validation accuracy.

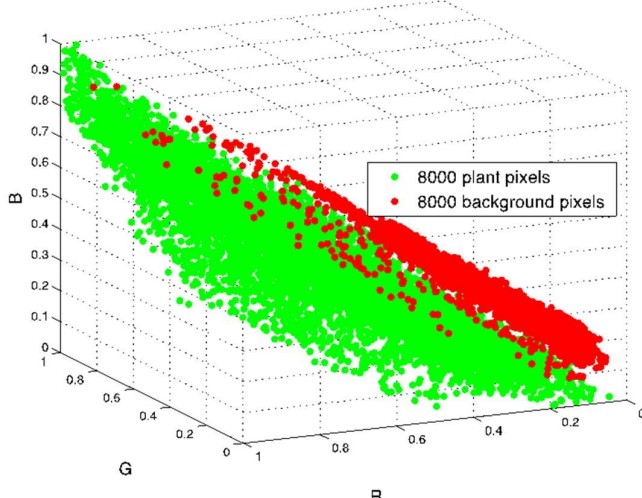

**Figure 7.** Scatterplot of {r, g, b} features of plants and background pixels.

### *2.3. Classification Evaluation*

To evaluate the classification accuracy, false negative and false positive rates of plant pixels were calculated. False negative is defined as a plant pixel that has been classified as a background pixel. A false positive is defined if a background pixel has been classified as a plant pixel. Five-thousand pixels in each category were chosen as ground truth data and compared with the segmentation result. The plant ground truth data were selected at leaf scale instead of plant scale because the canopy density does not provide total coverage of the soil or background. Since the SVM package already provided classification accuracy, the segmentation accuracy was evaluated only for perennial peanut at 9, 12, 15, and 18 m and arborvitae at 6, 12, and 22 m.

### *2.4. Counting Scheme*

### 2.4.1. Hypothesis

Although separating plant canopies with watershed algorithm [27] or detecting the circular shape with Hough transformation [28] is possible for cases in which the canopies slightly overlap (Figure 6). However, individual identification of compact-placed nursery containers (upper left block of Figure 2) is a great challenge. Canopy area mapping is a good alternative to estimate plant count. The hypothesis of the canopy area mapping method is that canopy areas of plants do not vary much. To validate the hypothesis, the distribution of canopy areas was estimated. If area information is easy to estimate based on a certain parameter (for example, area could be calculated if diameter is available for plants with circular shape), the distribution of plant area on the ground will be shown. However, similar to juniper (Figure 8), the ground leaf area is difficult to determine. The alternative is to find the distribution of plant area in the image (total pixels of each plant) based on the segmentation method. Arborvitaes, juniper, and rose placed on fabric were chosen as an example on which to check the distribution of the canopy area.

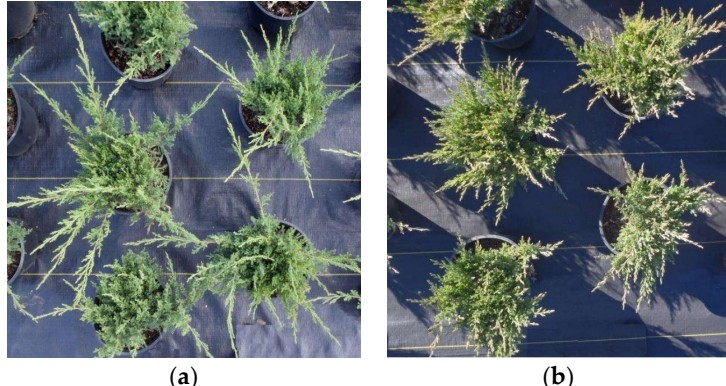

(**a**)                                                    (**b**)

**Figure 8.** Example of selected juniper: (**a**) with irregular shape; (**b**) with regular shape.

Arborvitae has a circular shape. Therefore, the distribution of the ground canopy area was estimated. There are nine sets of arborvitae. For each set, the shoot diameter of four corner plants and one center plant were sampled, and shoot diameters at two directions perpendicular to each other were measured. The diameter of each plant (d) was calculated by taking the average value of the two diameter measurements. The canopy area of each plant was calculated by $\pi \times \left(\frac{d}{2}\right)^2$. For juniper and rose, canopy areas in the images will be used instead due to the difficulty to estimate the ground canopy area.

### 2.4.2. Counting

Conceptually, the goal of the MATLAB algorithm is for a grower to use only a small number of container-grown plants to obtain a canopy area estimate, which can then be utilized to train the algorithm to provide an accurate inventory estimate of large plant blocks comprising container-grown plants. In this study, four separate plants were placed outside the region of interest (ROI). The classification result obtained from the proposed method was converted into a binary image: pixels with digit number 1 (white pixels) representing a plant and pixels with digit number 0 (black pixels) representing the background. Data was then used to calculate the average canopy area (A) in the training set and propagate the canopy area (A) in the training set to the ROI: Taking A as a basis, the number of white regions ($k_0$) with area smaller than $0.5 \times A$ was determined. Each region contains 0 plants. The number of white regions ($k_1$) with area in the range of $0.5 \times A$ and $1.5 \times A$ was determined. Each region contains 1 plant. Then the number of white regions ($k_2$) with area in the range of $1.5 \times A$ and $2.5 \times A$ is determined. Each region contains 2 plants. The process continues until all the white regions are included. The total count is the sum of plants in each region $0 \times k_0 + 1 \times k_1 + 2 \times k_2 + \ldots$ The flowchart of this process is shown in Figure 9. The counting accuracy is evaluated by Equations (2) and (3).

$$Accurary\ (i) = \frac{|average\ algorithm\ count(i) - ground\ count(i)|}{ground\ count(i)}, \tag{2}$$

$$Overall\ Accuracy = \frac{\sum_{i=1}^{n} Accuracy(i)}{n}, \tag{3}$$

where n is the number of replication.

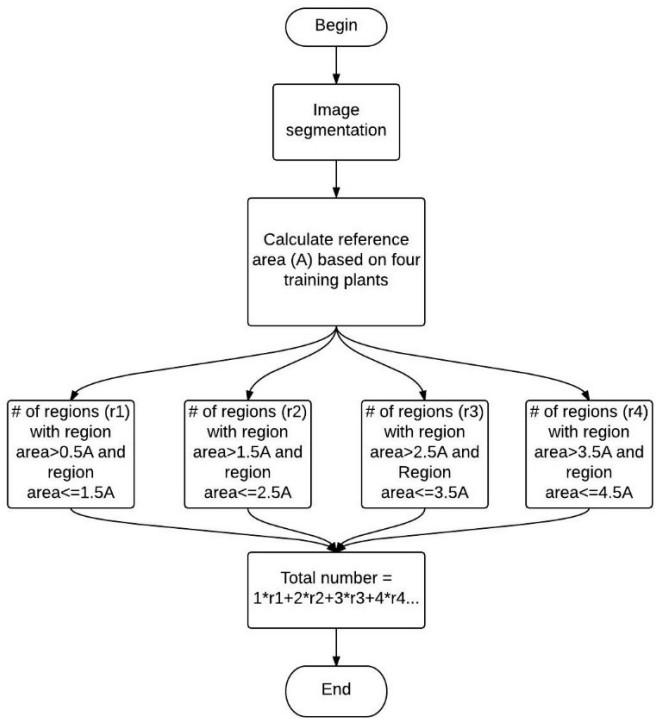

**Figure 9.** Flowchart of counting process.

The hypothesis of this counting method is that the canopy area does not vary too much. The reason is that growers tend to place the same species at the same growing stages in a certain region is that it will expedite the regulation and shipping process. Most plants with same growing stages will have similar canopy areas since they have same irrigation and weather conditions. Plants canopy areas vary; however, the algorithm has an integrated tolerance (0.5*base area).

## 3. Results and Discussion

### 3.1. Classification Results

To evaluate the classification accuracy, classification results for perennial peanut and Fire Chief[TM] arborvitae are listed in Tables 1 and 2, respectively.

**Table 1.** False negative and false positive rates of perennial peanut classification at each height.

| False Negative | False Positive | Height (m) |
|:---:|:---:|:---:|
| 3/5000 | 0/5000 | 9 |
| 0/5000 | 0/5000 | 12 |
| 3/5000 | 0/5000 | 15 |
| 0/5000 | 0/5000 | 18 |

**Table 2.** False negative and false positive rates of Fire Chief[TM] arborvitae at each height.

| False Negative | False Positive | Height (m) |
|:---:|:---:|:---:|
| 1/5000 | 0/5000 | 6 |
| 6/5000 | 0/5000 | 12 |
| 32/5000 | 0/5000 | 22 |

Table 1 showed that there were no background pixels that were misclassified as plants, while a few plant pixels were misclassified as background pixels. In addition, higher segmentation error at higher altitude due to the decrease of spatial resolution did not occur as expected. At the altitudes of

9 and 15 m, three out of 5000 plant pixels were misclassified as background. At the altitudes of 12 and 18 m, there were no misclassified pixels. The reason for this may be because the gap of component $(2 \times g - r - b)$ between the plants and background is large enough so that resolution is not a dominant factor in deciding the segmentation accuracy. In summary, the observed segmentation accuracy was >99%.

From Table 2, it is clear that the false negative rate increased as height off the ground increased. This is consistent with the assumption that the decreases in the spatial resolution will result in lower classification accuracy. However, the overall accuracy is >99%.

The evaluation of the flowering plant classification results is based on the LIBSVM tool. The grid search method found that the (C, $\gamma$) parameter pair of (16, 8) was with the highest cross-validation accuracy of 97.92% and was used to build the classifier for all training data. Furthermore, the classifier was used to predict test instances with unknown labels. The classification accuracy for the test data was 97.55% (3902/4000). Then, the built classifier was applied to the collected image, Figure 10 provides an example of the original image and binary segmented image.

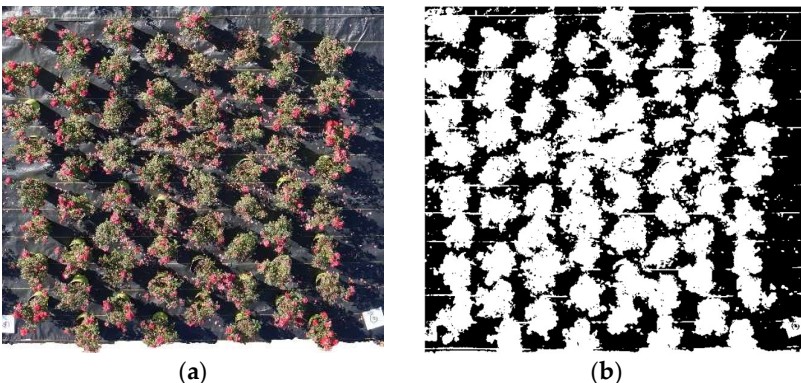

(**a**)　　　　　　　　　　　　　　　　　(**b**)

**Figure 10.** Example of segmentation of flowered rose from background: (**a**) original image; (**b**) segmented binary image.

From the comparison between the original and segmented image, it can be observed that leaves and flowers are successfully extracted from the background and shadows of the plants. Even the petals falling on the fabric are detected. The yellow grid lines on the fabric are also classified as plants. This is because the leaves (green) and flowers (red) were merged into one category. Their {r, g, b} feature space covered the feature space of yellow color. However, since the grid line is thin, it is removed by erosion operation with disk size = 2. The erosion operation will not only remove the grid lines, but also some information for leaves and flowers. However, the following dilation operation with the same disk size will bring back information for leaves and flowers but not for grid lines since they are already fully removed in the erosion operation.

*3.2. Validation of Counting Hypothesis*

In total, the ground canopy areas of 45 arborvitae were obtained. The distribution of the canopy area is shown in Figure 11.

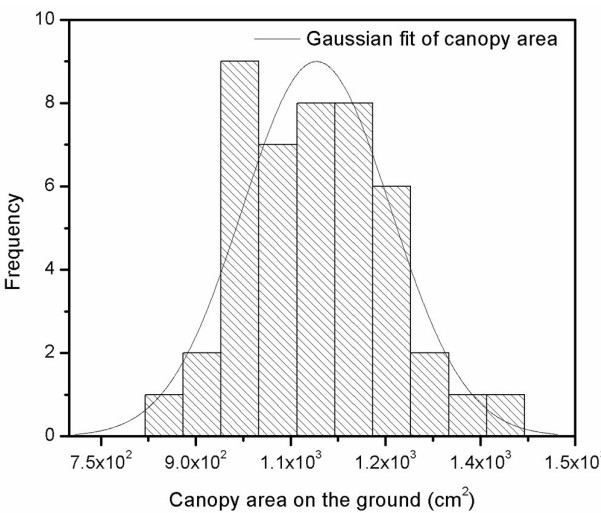

**Figure 11.** Distribution of ground canopy area of arborvitae.

The corresponding segmentation method for juniper and rose were used and canopy areas in the images were calculated. There were not many training plants available. Therefore, the boxplot with the mean and standard deviation of canopy area were created for juniper in Figure 12a and rose in Figure 12b.

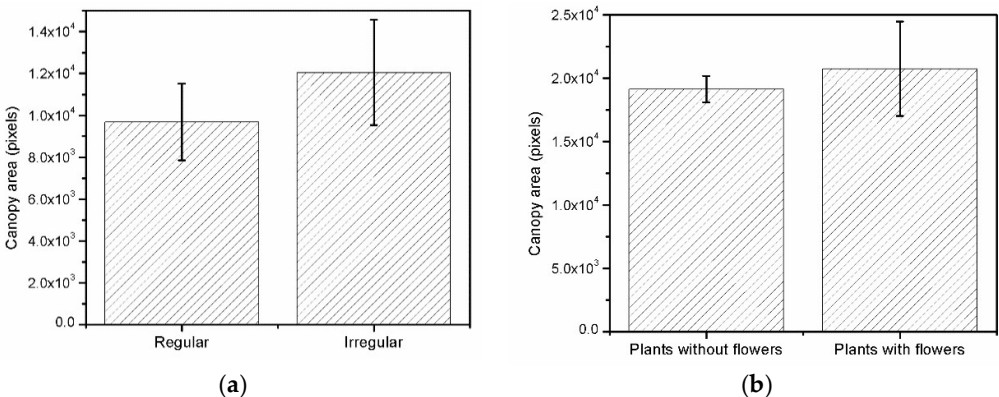

**(a)**          **(b)**

**Figure 12.** Boxplot of mean and standard deviation of canopy area in the image for regular and irregular juniper (**a**) and for rose with and without flowers (**b**).

The arborvitae canopy area on the ground fit a Gaussian distribution (1090.0 cm$^2$ ± 117.4 SD) with small variance which indicates that most of the canopy areas fall into a "narrow" range. The boxplots of the mean and standard deviation of the canopy areas for irregular plants (12,050 pixels ± 2519 SD), regular plants (9682 pixels ± 1837 SD), plants with flowers (20,478 pixels ± 3732 SD) and plants without flowers (19,153 pixels ± 1035 SD) indicated 5–20% variance of canopy area. For the same species or cultivar, management practices and weather conditions, we do not expect a large variance in canopy areas. The highest observed variance was 20% among our 5 different datasets. As mentioned earlier, the developed algorithm has an integrated tolerance (0.5*base area). Thus, it could help to mitigate some errors induced by canopy area variance.

### 3.3. Counting Results

### 3.3.1. Dataset 1. Effect of Height and Container Interval for Perennial Peanut

The plant blocks for some treatment pairs [container separation (cm), flight altitude (m)] were not included in the image due to the shift of the capturing system. These treatments include (0, 15), (3, 15),

(8, 15), (0, 18), (3, 18) and (8, 18). In addition, only seven rows (70 containers) were included in the image for treatment (13, 18). The counting accuracy for all available treatments is shown in Figure 13.

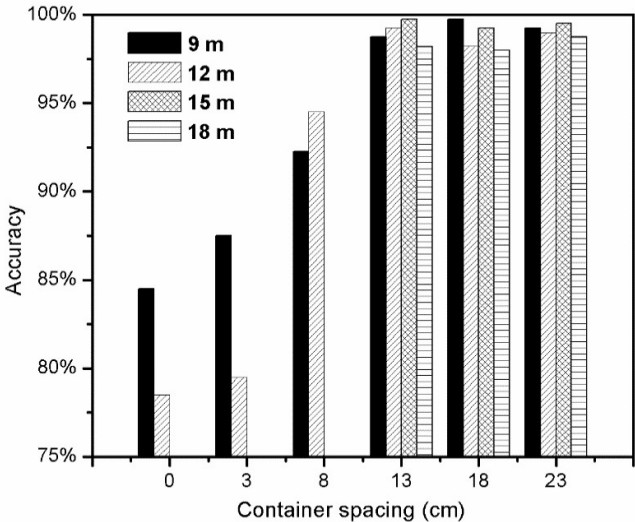

**Figure 13.** Counting accuracy for perennial peanut.

Figure 13 shows that the overall accuracy for container separations of 0 and 3 cm was >78%, whereas the accuracy for container separation of 8 cm was >92%, and for container separations of 13, 18, and 23 cm, accuracies increased to 98%. Therefore, counting accuracy increased as container separation increased from 0 cm to 13 cm. There was no obvious trend of counting accuracy when the canopy separation increased from 13 cm to 23 cm. Counting accuracy was only decided by the distribution of the canopy area since the canopies were not touching. The amount of container separation was not a key factor. In the case of 8 cm, where the canopy was touching but not overlapping, there was canopy interaction between adjacent plants. Thus, accuracy was not as good as when canopies were separated, but remained much higher than the case in which canopies are overlapped. When canopies are overlapped, in the case of the container separations 0 and 3 cm, there exists an underestimation of plants. This is because the reference area is much larger than the canopy area of container plants in the squeezed region of interest. If the reference area is taken as a basis and used for estimation, the count will be underestimated. Since the count data is not complete at the altitudes of 15 and 18 m with canopy separations of 0, 3 and 8 cm, it is impossible to reach a conclusion on the effect of altitude on the accuracy. However, when canopy separation is 13, 18, and 23 cm, capturing images at 15 m would be optimal.

### 3.3.2. Dataset 2. Sparse Container Block

The sparse block contains 80, 60, 40 and 20 ground containers. The experiment was replicated four times within each condition. A 95% confidence interval of four algorithm counts was used to evaluate the accuracy of algorithm for each missing condition. The results are listed in Table 3. A 95% confidence interval was reported for all count conditions, which indicates that the algorithm count is not significantly different from the ground count ($p = 0.05$).

**Table 3.** 95% confidential interval of 4 algorithm counts for each missing condition.

| Ground Count after Missing | 95% Confidential Interval |
|---|---|
| 80 | 75, 80 |
| 60 | 59, 62 |
| 40 | 37, 42 |
| 20 | 20, 20 |

To better observe the change in counting accuracy when the density of plants varied, the counting accuracy of the original 100 plants at 12 m spaced at 8 cm was also included. Counting accuracy for each condition is listed in Figure 14.

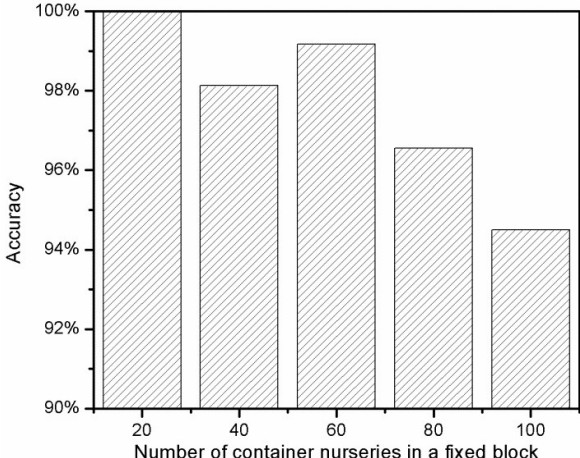

**Figure 14.** Counting accuracy for different sparse blocks.

The overall trend shows that the more plants removed from the block, the higher the accuracy. The only outlier was when there were 40 nursery containers remaining. The four replicated ground counts for this condition are 40, 40, 37 and 40. The relatively error resulting in low accuracy comes from the singular algorithm count of 37. The authors hypothesize this is a result of variation from the sun. The overall trend is a consequence of a more clearly defined canopy because more spacing results in less interaction between plants. The overall accuracy for all the missing case is >96%. This indicates that the algorithm could be applied for practical use when containerized plants are removed because of selling, shipping, or mortality.

### 3.3.3. Dataset 3. Effect of Height, Canopy Interval and Ground Cover for Fire Chief™ Arborvitae

To evaluate the effects of different variables and their combinations on the counting accuracy, a three-way analysis of variance was performed in JMP Pro 12. Results showed that spacing and the interaction of the background and spacing have a significant effect on the counting accuracy ($p = 0.05$). A Tukey honest significant difference (HSD) test showed that accuracy at spacing of $-5$ cm was significantly different from that of 0 and 5 cm (Table 4). No significant effect of heights on counting accuracy was found. It could be observed that the difference between the highest and lowest accuracy with different heights cannot exceed 2% with fixed canopy spacing. This indicated that image resolution is not a dominant factor on counting accuracy. The background had no effect on accuracy as well. Although fabric background has a larger contrast with plants than gravel, the counting accuracy has no significant difference.

**Table 4.** Tukey honest significant difference (HSD) test of effect of spacing on counting accuracy.

| Level | | | Least Square Mean |
|---|---|---|---|
| 5 | A | | 0.94 |
| 0 | A | | 0.93 |
| $-5$ | | B | 0.74 |

Means with the same letter (A and B) are not significantly different based on the Tukey HSD test (Q = 2.38, $p = 0.05$).

Background and spacing have an interaction effect on the accuracy that can be clearly seen in Table 5. For the black fabric background, the overall accuracy for Fire Chief™ arborvitae that was complete separated (canopy interval 5 cm) was >97.8%. When touching (interval 0 cm),

the accuracy was approximately 94.2%, while the accuracy for overlapping canopy (interval −5 cm) was approximately 72.0% (Figure 15). This was consistent with the results reported in perennial peanut. For the gravel background, the accuracy for touching and separate are 91.7% and 89.8%, while the accuracy for overlapping is approximately 75.8% (Figure 16). The possible reason that the separated case fails to overweigh the touching case is that the poor segmentation between plants and the gravel background is due to the similarity between plants and gravel. The interval space in the separate case between adjacent plants is a source of error.

**Table 5.** Tukey HSD test of spacing and background on counting accuracy for Fire Chief[TM] arborvitae.

| Level | | | | Least Square Mean |
|---|---|---|---|---|
| 5, Fabric | A | | | 0.98 |
| 0, Fabric | A | B | | 0.94 |
| 0, Gravel | | B | | 0.92 |
| 5, Gravel | | B | | 0.90 |
| −5, Gravel | | | C | 0.76 |
| −5, Fabric | | | C | 0.72 |

Means with the same letter (A, B and C) are not significantly different based on the Tukey HSD test (Q = 2.91, $p$ = 0.05).

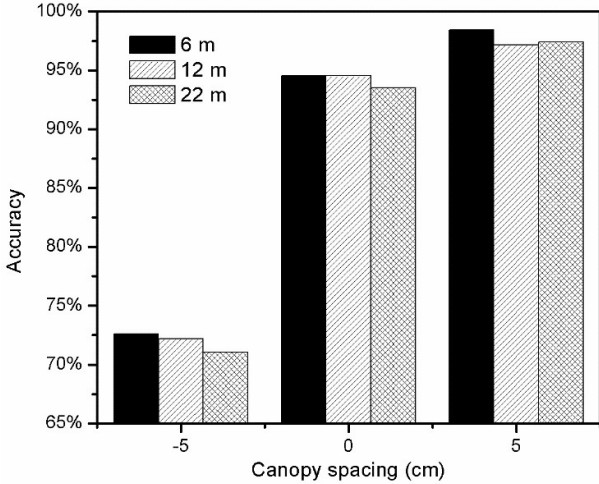

**Figure 15.** Counting accuracy for Fire Chief[TM] arborvitae on black fabric with different canopy spacing and at different capturing heights.

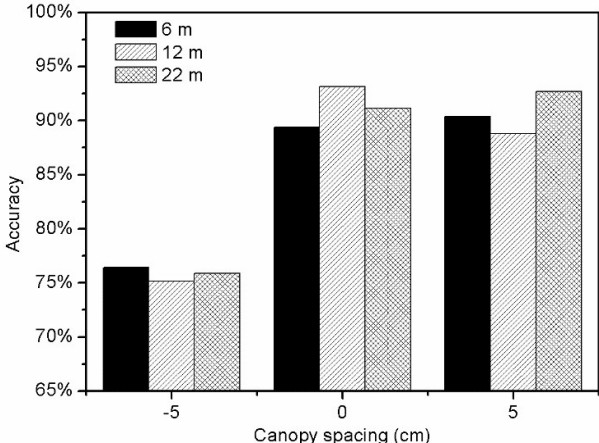

**Figure 16.** Counting accuracy for Fire Chief[TM] arborvitae on gravel with different canopy spacing and at different capturing heights.

Since all the counting results are based on the reference area, it is valuable to check the performance of the algorithm in one standard deviation range of the reference area. Interval counting was applied to each block with two other base areas, mean (training areas) − std.dev. (training areas) and mean (training areas) + std.dev. (training areas). This will provide a counting range [low, high]. The [low, high] range could give the growers a rough inventory number. In Figure 17, the average counting gaps (high-low) were listed at different heights and at different spacing. The average gap for sets with 0 cm canopy spacing is 11, while the average gap for sets with 5 cm and −5 cm canopy spacing are 5 and 10, respectively. The sets with touching canopies and overlapped canopies have more fluctuation with variation of the reference area than the squeezed block. This is a result of erosion and dilation operations. For separate cases, the effect of the morphology operation is smaller compared to that of the touching and overlapping cases. It is clearly observed that the counting gap at 12 m is minimal in all spacing cases. This indicates that the algorithm behaves more stable at 12 m when there is variation in the reference area.

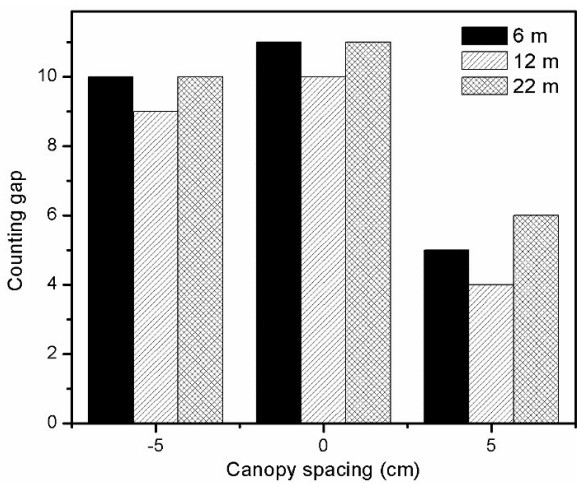

**Figure 17.** Counting gap (high-low) according to shift of reference area.

### 3.3.4. Dataset 4. Effect of Nursery Canopy Shape and Effect of Presence of Flowers

This experiment included two runs. In each run, two treatments were replicated five times in a randomized block design. In total, there are 10 counting results for each treatment case, respectively.

The accuracy for regular plants was 98.13% while the accuracy for irregular plants was 94.38%. This indicated that the algorithm performs better when the canopy shape is regular. The possible reason is that the standard deviation of the canopy area of irregular plants is larger than that of regular plants. The counting algorithm is built based on the assumption that the variance of canopy area is small across the ROI. The more variance the canopy area has, the less accurate the algorithm.

The accuracies for plants with and without flowers were 96.88% and 97.66%, respectively. The results indicate the same SVM model, and the interval counting scheme works well when the ROI and training plants are within the same category (plants with and without flowers).

### 4. Conclusions

A counting algorithm in MATLAB was developed to assist with inventory management of containerized nursery crops from aerial images. Research utilized four taxa (peanut, arborvitae, juniper, rose) to create multiple real-world scenarios to both develop and validate the algorithm that includes image classification and a count estimate.

For image classification, two representative taxa were selected which included green foliage and flowering plants. Different classification methods were developed for each scenario. Classifiers were built based on color information in which the background cover (black ground cloth) remained the same. In the future, if there are plants of similar color to our representative plants,

the same classification method could be used and could serve as a database for future classification. The classification accuracy on a pixel-based level was evaluated. The ground truth data was manually chosen within the image. For perennial peanut, the classification accuracies at different altitudes were evaluated. The false negative rate was approximately 2/5000 with no false positive pixels. For rose, the overall accuracy of SVM segmentation was 98%.

A counting estimate method utilizing training plants outside the ROI was developed. The counting method was based on the canopy area in the image (pixels). The algorithm was tested in different experiments. Experiments were performed to test the effect of the altitude at which data was collected, canopy spacing, and background cover on counting accuracy. Results show that canopy spacing has a significant effect on counting accuracy. Overall, counting accuracy increased when canopy spacing increased. The collecting altitude was not a dominant factor. The interaction between canopy spacing and background cover also had a significant effect on counting accuracy.

Sparse blocks of nursery containers with 100, 80, 60, 40, or 20 containers within a fixed area were created to test the performance of the algorithm when the quantity of plants changed. The accuracies for all cases were greater than 94%. The overall trend shows the lower the density, the higher the counting accuracy. These results could be practical in a nursery setting since plant densities are constantly changing when plants are removed from the production block due to shipping, selling, and mortality.

The counting accuracy was compared for irregular-shaped and regular-shaped plants, as well as plants with and without flowers. Results show that the counting accuracy for regular plants (98.13%) is higher than that for irregular plants (94.38%), and the counting accuracy for plants without flowers (97.66%) is higher than that for plants with flowers (96.88%). This proves that the counting scheme could work for plants with different shapes and presence of flowers.

In summary, this research has proved that aerial imagery is a fast and stable way to conduct in-field container nursery inventory management. Future work should be focused on evaluating the algorithm under different light conditions. Image normalization may be needed to make the index threshold more generalized. Creating a commercial product based on the developed algorithm that allows growers to customize the parameters based on their needs could be another focus.

**Author Contributions:** Y.S. developed the methodology, completed the data analysis, and wrote the original draft. R.E. was the project administrator, helped provide the resources for this project, and also editing the manuscript. J.R. helped with developing the methodology and provided the resources for data collection as well as editing the manuscript. J.N.L. helped with the literature review and assisted with data collections. J.O. helped with developing the methodology, data collection, and editing the manuscript.

**Funding:** This research was partially supported by the United States Department of Agriculture—National Institute of Food and Agriculture under Award #2013-67021-20934 and a fund from Oregon Department of Agriculture (ODA)/Oregon Association of Nurseries (OAN).

**Acknowledgments:** We would like to thank Julie Brindley and Sherrie Buchanon for assisting in data collection and Joe Mari Maja for assistance with the UAV.

**Conflicts of Interest:** The authors declare no conflict of interest.

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
