# Peer review of "Applications of High-Resolution Imaging for Open Field Container Nursery Counting"

_remotesensing, doi:10.3390/rs10122018_

Round 1

Reviewer 1 Report

Dear authors

I would like first to say that the paper is well written, easy to read. I have noticed some mistakes which can be found in the file attached.

You paper is interesting but needs more details on the methods used and on your real contribution to the domain. Moreover, since you do all your experiments directly outside, the problem of the illumination begins crucial and I am a little bit surprised that it is just mentioned in the conclusion.

I have however some other remarks concerning the experiments done, the data analysis and more generally on the methods used.

1/ when you mention the data collection, why didn't you use a UAV ?

2/ Line 129: How do you mark the center location of the 10*10 block ? Manually ? Do you need to do that for each experiment ? If yes, this means that the method proposed is not automatic, so what about the calculation time ?

3/ Line 133: what about the image format ? The resolution appears interesting (20 Mpixels) but what is also important is the possible compression of the data due to the image format.

4/ I don't really understand the interest of the dataset 2 ? May you give more explanations ?

5/ Line 148: you mention 3 plant densities: are these values generally used in the market ?

6/ Lines 169/170: you mention that you remove manually the flowers of the roses. A grower should also do this for other plants than roses ?

7/ For the figure 1, the discrimination between plants and background is easy since the threshold is well defined. What is the result if the background is not black or with some textural information close to the plant ones ? Why not to use an automatic method based for example on an optimization of the Otsu method ?

8/ Lines 213/214: How do you establish the novel index ? No theory is provided in the paper. Did you test several possibilities to find the best one ?

9/ For the figure 4: why not to have tested a Hough Transform ?

10/ Figure 8: If you want only to count, you don't need to have an accurate segmentation no ?Do you think you can envisage to have two different methods ?

11/ Lines 365/367: if you use an erosion operator, even if it is small, you will also lose some leaves/flowers information

12/ Lines 382/388: I am not agree with the authors concerning the small variance in canopy area. It is small for the "plants without flowers" but around 20% for the three others !

13/ the references are for the majority too old !!

14/ Why not to have cross the different dataset to do multivariate analysis ?

Author Response

Please see attachment. Thank you.

The reviewer comments are in black and our responses are in red

Reviewer 2 Report

In the manuscript entitled “Applications of High Resolution Aerial Imaging for Open Field Container Nursery Counting” the authors propose an algorithm for nursery species inventory. Such studies are particularly important for promoting low cost, efficient and a lot less labor intensive remote sensing techniques to nursery community. Overall paper and results are very interesting and the authors were very careful in defining the work methodology, trying to cover the various parameters capable of influencing the results. However, some doubts/issues need to be addressed before recommendation for publication.

Below, I include my recommendations:

1. It is stated that there exist thousands of nursery species which make impossible to develop a method working in all cases. I agree, however, a better justification about the reasons behind the selection of these specific species should be provided? Are they representing the most significant portion of the business? Are they easy to classify/detect? Etc.

2. I suggest to reorganize the introduction section. It mixes the revision of the state of the art with the paper goals and then more state of the art regarding studies that only marginally relate to the work. The authors say that fruits counting strategies cannot be applied for nursery, so mentioning that several methods/studies were developed to count fruits (references), however for [all these reasons] they cannot be applied in this specific case.

3. From lines 85 to 90, the authors state that previous approaches use expensive software. But they immediately say that they use matlab which is not cheap at all (of course executable can be done, etc.). Please try to change the sentence focusing on the add value of the proposed method.

4. In section 2.1. it is described the equipment used for data collection. It would be interesting to add a photo of the equipment both boom and drone in operation.

5. The whole experiment was carried out in two days. I am skeptical regarding this because it would be not possible to test all the parameters that can influence the results. The authors comment on this when they had to analyze shadows. What kind of improvements would be expectable if the experiment had been carried out during a longer period?

6. Please give more details about flight missions: simple/double grid, overlaps, time of flights, etc. This would help understanding some problems/differences in the acquired images.

7. Image segmentation was simple because of the used background. However, what would be the most constrains of the method if used in other kind of background? In lines 456 to 459 the authors recognize that limitation.

8. In subsection 2.2.2 it is explained that the k value was set via a trial and error process. So this has to be done in any situation? That would limit the effectiveness of the method.

9. The method was applied to different species separately. It would be interesting the see the method’s behavior applied to a scene/scenario with different species together.

10. The fact that a reference area has to be measured is from my point of view one of the major limitation of the method. Wouldn’t be possible to use image processing techniques (e.g., shapes intersections, etc.) to automate the process? Can you comment on this?

11. In line 411, the authors say that “the count data is not complete at the altitudes of 15 and 18 m with…” It would allow to reach a conclusion on the effect of altitude on the accuracy, so I don’t understand why the experiment wasn’t repeated…

12. In lines 431 to 433 the authors indirectly agree in my comment #5.

Author Response

    Please see the attached file for the details responses

Round 2

Reviewer 1 Report

Dear Authors

I would like to thank you for the efforts on the answers.

The introduction has been greatly improved to better understand the objectives of your research. You need also to better explain the differences between your research and the other ones. New figures have been added and the equations have been numbered.

However I have again some remarks :

1/ the data collection are acquired with a boom not with a UAV. So take care about the title !

2/ I don't understand exactly how the stitching is done ? (with which corrections)

3/ the normalization is not done for your experiments but in general the acquisitions may not be done under the same environmental conditions

4/ you don't tested the Hough transform so you explanations is not sufficient to my opinion

5/ finally for the variance, I am not convinced with the 20% found !

Reviewer 2 Report

I appreciate the authors' effort to improve the quality of their work. I believe the manuscript is ready to be published.

Author Response

Thank you for your valuable comments.